# Optimizing Early Neonatal Nutrition and Dietary Pattern in Premature Infants

**DOI:** 10.3390/ijerph18147544

**Published:** 2021-07-15

**Authors:** Cornelia Wiechers, Wolfgang Bernhard, Rangmar Goelz, Christian F. Poets, Axel R. Franz

**Affiliations:** 1Department of Neonatology, University Children′s Hospital, Eberhard Karls University, Calwerstr. 7, 72076 Tübingen, Germany; wolfgang.bernhard@med.uni-tuebingen.de (W.B.); rangmar.goelz@med.uni-tuebingen.de (R.G.); christian-f.poets@med.uni-tuebingen.de (C.F.P.); Axel.Franz@med.uni-tuebingen.de (A.R.F.); 2Center for Pediatric Clinical Studies, University Children′s Hospital, Eberhard Karls University, 72076 Tübingen, Germany

**Keywords:** preterm infant, nutrition, enteral feeding advancements, growth

## Abstract

Providing adequate amounts of all essential macro- and micronutrients to preterm infants during the period of extraordinarily rapid growth from 24 to 34 weeks’ postmenstrual age to achieve growth as in utero is challenging yet important, since early growth restriction and suboptimal neonatal nutrition have been identified as risk factors for adverse long-term development. Along with now well-established early parenteral nutrition, this review emphasizes enteral nutrition, which should be started early and rapidly increased. To minimize the side effects of parenteral nutrition and improve outcomes, early full enteral nutrition based on expressed mothers’ own milk is an important goal. Although neonatal nutrition has improved in recent decades, existing knowledge about, for example, the optimal composition and duration of parenteral nutrition, practical aspects of the transition to full enteral nutrition or the need for breast milk fortification is limited and intensively discussed. Therefore, further prospective studies on various aspects of preterm infant feeding are needed, especially with regard to the effects on long-term outcomes. This narrative review will summarize currently available and still missing evidence regarding optimal preterm infant nutrition, with emphasis on enteral nutrition and early postnatal growth, and deduce a practical approach.

## 1. Background

Feeding preterm infants is challenging because their nutritional needs are higher than those of term infants. This is due to their 4-fold higher physiological growth rate during this developmental period—the last trimester of fetal development—which occurs in the neonatal intensive care unit (NICU) rather than in utero. This phase is characterized by extraordinarily rapid, exponential growth from 24 to 34 weeks’ postmenstrual age (PMA), and the acquirement of notable amounts of lean body mass, fat and reserves of micronutrients. Remarkably, between 24 and 40 weeks PMA, adipose tissue increases 80 fold, water 4 fold and lean body mass solid matter 11 fold [1].

Achieving growth rates of all body compartments and organs that are similar to those in utero should be the primary goal of preterm infant nutrition [2]. Despite more intensive feeding strategies for preterm infants in recent years, growth failure remains a common problem in very preterm infants during their postnatal hospitalization [3,4] and is associated with impaired neurocognitive outcomes [5,6,7,8]. Furthermore, premature infants are at an increased risk of cardiovascular disease and insulin resistance in adulthood [9,10,11]. The underlying link between preterm birth and later metabolic alterations is still poorly understood, and early-life growth restriction, as well as excessive catch-up growth after initial growth failure, have been reported [8,12]. Conceivably, the inadequate postnatal growth of preterm infants may have deleterious “programming” effects on metabolic health similar to those of intrauterine growth restriction in term infants [13,14], occurring during the same period of development.

Postnatal growth of preterm infants in the NICU is the result of a complex interaction of various factors, such as neonatal morbidity and inflammation preventing anabolism, increased respiratory work causing increased energy requirements and, of course, nutrition and the immaturity of the gastrointestinal tract [15]. Therefore, providing adequate macro- and micronutrients to preterm infants and achieving growth similar to that in utero is challenging [2].

In the last decades, an increasing number of studies has been carried out regarding the nutrition of premature infants [16,17,18,19,20], but knowledge remains limited, e.g., concerning the optimal macro- and micronutrient intake through parenteral nutrition (both in the first week after birth and thereafter), the best way to achieve full enteral feeding, or indications for and the optimal composition of a breast milk fortifier.

Furthermore, long-term outcome data in infants following various nutritional interventions are often lacking. This review summarizes the perspective of a level 3 NICU with a focus on very early enteral nutrition and its repercussions on other aspects of neonatal nutrition (Table 1 and Table 2). Therefore, we searched PubMed and the Cochrane library for each of the topics addressed above to provide a narrative review of current evidence, open questions, as well as controversial aspects.

## 2. Parenteral Nutrition

During the first postnatal days, complementary to enteral feeding, parenteral nutrition is an integral part of preterm infant care, bridging the period until full enteral feeding is established. Although neonatal parenteral nutrition has been established since the late 1960s [38], and has improved considerably since, evidence on the optimal composition of macro- and micronutrients is limited, yet intensively discussed. Several studies and systematic reviews have shown improved short-term growth and a shortened time to regain birth weight using neonatal parenteral nutrition. However, according to randomized controlled trials, its long-term benefit for metabolism and neurological development is still unclear [39,40,41,42]. Nevertheless, data on associations between higher nutrient intake and improved growth suggest that parenteral nutrition in the first postnatal weeks is likely to improve cognitive outcomes [5,42,43].

Therefore, initiating parenteral nutrition immediately after delivery is recommended for preterm infants (Table 1) [27]. Since 2018, the guidelines of the European Society for Paediatric Gastroenterology, Hepatology and Nutrition (ESPGHAN) recommend an amino acid supply of at least 1.5 g/kg/d for the first postnatal day, increasing to 2.5–3.5 g/kg/d from postnatal day 2 onwards [21]. To provide a rich source of energy at low volume, intravenous lipid emulsions are an indispensable component of neonatal parenteral nutrition. These can be started shortly after birth but should not exceed 4 g/kg/d [22]. In a systematic review including 29 studies with >2000 infants, no benefit of new lipid emulsions including fish oil, compared to conventional soybean oil-based lipid emulsions, was found for the prevention of cholestasis, growth, mortality, retinopathy of prematurity and bronchopulmonary dysplasia (BPD) [41]. However, according to the current ESPGHAN recommendation, the latter emulsions should not be used for more than a few days in term and preterm neonates, as pure soybean oil may provide a less balanced nutrition than compound fat emulsions (e.g., soybean/olive with or without fish oil) [22].

A parenteral glucose supply should start at 4–8 mg/kg/min, and avoid overfeeding or excessive glucose load by regular blood glucose measurements [23]. The maximum endogenous glucose production, as well as the glucose oxidation rate, which both are approximately 7–8 mg/kg/min (10–11.5 g/kg/day) in preterm infants, should not be exceeded, at least not initially. In addition, though effects on growth have mostly been studied for macronutrient intakes, an adequate micronutrient intake is also necessary for tissue, particularly parenchyma accretion [24,25]. In line with this, early and enhanced postnatal parenteral nutrition is associated with increased electrolyte requirements, particularly concerning phosphate and potassium, to meet the increased anabolism of parenchymal tissues [24,44].

Standardized parenteral nutrition solutions and computerized prescriptions are recommended to improve patient safety [28]. Therefore, the authors practice nutrition with standardized in-house “parenteral starter solutions” starting in the first postnatal hour and containing 3.5 g/kg amino acids and 4.2 mg/kg/min glucose, as well as small amounts of calcium, phosphate, and sodium. With an additional standardized fat emulsion including fat- and water-soluble vitamins, a fat intake of 2.5 g/kg/d is achieved from the first day of life onwards.

## 3. Transition from Intravenous to Enteral Sources

Prolonged parenteral nutrition is associated with cholestasis, thrombosis, infectious and metabolic risks [29], partly due to an inadequate composition of existing products. Hence, meeting nutritional needs through full enteral nutrition is a general goal. Remarkably, the intestine of an extremely premature infant can already digest, tolerate and metabolize human milk. In utero, amniotic fluid with its bioactive peptides not only plays an important role in fetal gastrointestinal development [45], but animal data show that up to 14% of intrauterine nutrient requirements are supplied and absorbed prenatally through the intestine [46,47].

Nevertheless, evidence-based recommendations for the transition from intravenous to enteral nutrient supply are lacking, and the optimal rate of enteral feeding advancements in preterm infants is unclear [16,29,48,49]. Frequently, there are concerns that rapid enteral feeding advancements may cause necrotizing enterocolitis (NEC). In a systematic review of 9 studies including 1106 preterm infants, however, no increased risk of NEC was found with early enteral feeding starting within 3 days after birth compared to a more delayed onset [50]. Current evidence suggests that accelerating enteral feeding volume in daily increments of 30 mL/kg does not increase the risk of NEC, death, or neurodevelopmental disability at 24 months in preterm infants [16,29,48,51]. Instead, growth rates similar to intrauterine trajectories can be achieved by rapid increases in enteral feeding volume and by achieving full enteral nutrition within 5–7 days of birth even in extremely low gestational age neonates [48,49]. However, even if weight gain along intra-uterine trajectories is achieved, very preterm infants still show insufficient lean body mass growth and an increased fraction of body fat at term-equivalent age [52,53]. This can be interpreted as a lack of essential nutrients required to achieve parenchymal and muscle mass growth as in utero [54].

Particularly during the transition from intravenous to enteral nutrition, “increased” gastric residues are detected during routine evaluation. The belief that increased gastric residuals may be predictive of NEC frequently results in withholding or delaying enteral feeding advancements [30]. Physiologically, however, 2–4 mL/kg of gastric residual fluid is regularly aspirated just prior to any scheduled feeding [55]. Similarly, gastric residual volume varies depending on patient position and feeding tube size and position, further limiting the clinical usefulness of this practice [56]. A recent systematic review showed insufficient evidence to support routine surveillance of gastric residuals with the intention to prevent NEC [30]. In a subsequent randomized controlled trial involving 143 preterm infants below 1250 g birth weight, no benefit was found for this practice [17]. Obviously, this small study was under-powered to assess the effect of such a practice on NEC rates, but the group that had no gastric residues determined showed an advanced enteral feeding pattern and higher feeding volumes by week 5 [17]. Furthermore, discarding gastric residuals (e.g., those showing a dark green color) results in a loss of bile acids and phosphatidylcholine from bile and digestive enzymes from the pancreas and enterocytes, all playing important roles in intestinal homeostasis, regulation and digestion [57]. Thus, based on current evidence, restricting monitoring of gastric residues to infants with symptoms of severe gastrointestinal dysfunction, such as emesis, abdominal tenderness, absent bowel sounds or bloody stools likely improves enteral nutrient supply. However, the safety of this approach remains to be proven.

## 4. Enteral Nutrition

Unquestionably, mothers’ own milk is the preferred source of nutrition for preterm infants because of its numerous short- and longer-term health benefits, such as protection against NEC, late-onset sepsis, and bronchopulmonary dysplasia (BPD), as well as improved neurodevelopment [32,58]. Additionally, early oral administration of colostrum provides immunological components, probably stimulating the immune system and protecting from inadequate bacterial colonization by lactoferrin, sIgA and other compounds [59,60]. Moreover, expressed breast milk contains high numbers of myeloid-derived suppressor cells, which may prevent excessive inflammatory reactions and enhance tolerance to food antigens [61,62].

However, mothers of preterm infants face a variety of barriers against breastfeeding. Expressing breast milk approximately eight times/day during the first two weeks after birth is the only proven way to increase the likelihood of achieving an adequate milk supply of more than 500 mL per day, necessary for subsequent exclusive or predominant breastfeeding. To achieve this, mothers must be encouraged to start manual and/or mechanical breast milk expression shortly after birth [63,64]. In addition to the emotional stress and concerns about the health of their baby [65], expressing breast milk 2–3 hourly, as recommended, is time consuming and needs to be included in a tight daily routine with kangarooing and, if present, caring for older siblings [66]. Appropriate prenatal counselling and postnatal support for a variety of systems (e.g., double electric, bedside and free home breast pumps for milk expression) by clinical staff in the NICU, peer support, skin-to-skin care, staff education and a lactation consultant should therefore be common practice [67].

When mothers’ own milk is available in insufficient quantity or is contraindicated (e.g., acquired immunodeficiency syndrome, chemotherapy), donor human milk (DHM) is the adequate substitute for preterm infant feeding, as recommended by all relevant societies [2,20,31,32,33]. In 2019, a systematic review of 12 studies, including 1879 infants <2500 g birth weight, showed that formula feeding, compared to predominantly non-supplemented DHM, resulted in improved weight gain, linear length and head growth, indicating an inadequately low nutrient supply through non-supplemented DHM [33]. However, formula feeding also resulted in a higher risk of NEC (typical risk ratio (RR) 1.87, 95% CI 1.23 to 2.85) [33] and was associated with a lower quantity of breast- compared to formula feeding at discharge [20]. DHM is not available in all NICUs and is considerably more expensive than formula (e.g., $15/100 mL from a US not-for-profit Human Milk Bank compared to $3/100 mL for preterm formula) [68]. By contrast, from a societal perspective, the total cost of providing DHM to preterm infants is equal to formula feeding, due to a reduced NEC rate [69]. In essence, it is important to reiterate that fresh mothers’ own milk is the first choice for feeding preterm infants, and that great efforts should be made to promote lactation, bridging the time to sufficient breast milk supply with (supplemented) DHM [31].

## 5. Fortification

Human breast milk is optimally designed for term newborns and infants, who double their weight within 4–6 months after birth. However, in line with the physiological intrauterine growth rate during the 3rd trimester, very preterm infants double their weight within 4–6 weeks. Thus, an increased supply of macro- and micronutrients is necessary for adequate growth. Requirements for energy, protein, (essential) fatty acids, minerals such as calcium and phosphate, as well as micronutrients like iron and vitamin D, to name a few, are higher than in healthy newborns. For additional constitutive components, like choline, increased requirements compared to current recommendations are debated as well [54]. All these nutrients are principally present in multi-nutrient human milk fortifiers (HMF), although their quantities are still controversial [54,70]. Hereby, intakes recommended by ESPGHAN in 2010 (energy: 110–135 kcal/kg/d, protein: 4.0–4.5 g/kg/d (<1 kg) and 3.5–4.0 g/kg/d (1–1.8 kg), carbohydrate: 11.6–13.2 g/kg/d, fat: 4.8–6.6 g/kg/d) should be achieved [2]. However, several recent studies indicate a ceiling effect for the beneficial effect of protein intake on growth at approximately 4.5 g/kg/d [18,71].

Although breastmilk fortification is practiced in most NICUs [34], evidence for its impact on long-term outcomes is remarkably sparse [72]. In a systematic review, multi-nutrient fortification of human milk vs. non-fortified human milk showed increased in-hospital growth rates for weight, head circumference and length without increasing the risk of NEC [72]. However, the limited follow-up data for post-discharge growth and neurodevelopment in later childhood show no benefit from fortification [72]. A recent meta-analysis showed that early fortification starting at 20–40 mL/kg/d of enteral feeds vs. late fortification (starting at 100 mL/kg/d) had little or no effect on short-term growth outcomes [73].

However, as breast milk has no uniform nutrient content and marked inter- and intra-individual variability exists [74], there are strategies for individualizing fortification to match the nutritional needs of preterm infants [34]. Individual fortification is performed by measuring the infant’s blood urea nitrogen (‘adjustable’ fortification) or the macronutrient content of breast milk using a milk analyzer (‘targeted’ fortification). In a recent review including 7 RCTs with a total of 521 participants, increased growth rates for weight, length and head circumference were found with moderate to low evidence following individualized compared to standard non-individualized fortification [35]. In 2021, a double-blind, randomized controlled trial was conducted in 103 preterm infants <30 weeks comparing standard versus targeted fortification with modular proteins, lipids, and carbohydrates [52]. The targeted fortification group had higher macronutrient intakes and higher average growth velocity across the first 21 days of intervention (21.2 ± 2.5 vs. 19.3 ± 2.4 g/kg/d). Not surprisingly, infants born to mothers with a low breast milk protein content showed the greatest benefit from targeted fortification regarding their weight at 36 weeks, length, head circumference, fat and fat-free mass [52]. Likewise, donor milk often contains low levels of protein, suggesting targeted fortification to improve growth [75,76]. However, data on the clinical benefit of individual fortification by adjusting breast milk macronutrients beyond short-term growth are sparse and inconclusive. A secondary analysis of a randomized controlled trial indicated that ‘adapted’ protein supplementation, by calculating breast milk protein content based on the duration of lactation, may be an easy and inexpensive alternative to ‘targeted’ protein supplementation for achieving protein supply on target in >95% of analyzed breast milk samples [77].

In recent years, discussions have addressed the question of whether multi-nutrient fortifiers (HMF) derived from human rather than bovine milk may further reduce the risk of NEC. However, the potential benefits of HMF derived from human milk have been insufficiently investigated, especially in comparison with feeding regimens without supplementary formula feeding, the latter already known to increase NEC rates. In 127 preterm infants <1250 g, an RCT using human vs. bovine milk-derived HMF in infants fed human milk failed to improve feeding tolerance [78], but was underpowered to assess effects on mortality and morbidity such as NEC. Moreover, concerns exist against the commercialization of human milk, as the milk used to produce the fortifier is no longer available as donor milk for very preterm infants. Other disadvantages are high cost, unequal access to these products in different countries, and the fact that the large volume of such liquid human milk-based HMF reduces the volume of expressed breast milk administered to the infants by up to 1/3. Therefore, the use of human milk-based HMF is currently not recommended by most committees and experts on pediatric nutrition [34,79].

## 6. Breast Milk–Acquired Cytomegalovirus Infection

Cytomegalovirus (CMV) reactivates in the lactating breast of up to 96% of CMV-seropositive mothers, i.e., in approximately 50–80% of all mothers of preterm infants [19,80]. It can cause severe sepsis-like symptoms with highly variable organ manifestations [80,81]. Postnatal transmission occurs in approximately 40% of infants <32 weeks’ gestation, with higher transmission rates the lower gestational age is [19,80]. In most cases, at least one of the following clinical signs is found: apnea and bradycardia, hepatosplenomegaly, hepatitis, pneumonitis, intestinal distention and altered laboratory parameters (including lymphocytopenia, neutropenia, thrombocytopenia, and elevated liver enzymes) [19]. Although most of these symptoms are self-limiting, CMV-related deaths have also been reported [81]. In addition, an increased risk of developing BPD has been described in cohorts of 385 extremely [82] and 2132 very low birth weight (VLBW) infants [83], as well as in a recent study involving >100 000 VLBW infants [84]. NEC also seems to be associated with postnatal CMV infection (pCMV). In numerous case reports and case series, pCMV could be identified in gut specimens [85,86]. A recent study enrolling 596 VLBW infants found an almost 3-times higher risk of NEC in CMV-positive than in CMV-negative infants [87]. An actual multicenter study comprising 304 VLBW infants with postnatal CMV observed significant associations with hearing and growth impairment, as well as a prolonged hospitalization (by 12 days), but not with NEC [88]. Principally, all organ systems, including the brain, can be involved in pCMV disease, as shown by autopsy findings [89].

Some controversy exists whether early pCMV has negative consequences for neurocognitive development [81,90,91]. In a recent cohort study involving 356 infants <32 weeks’ gestation [91], no negative impact on neurodevelopment until age six years was found in a subgroup of 49 CMV-positive infants (14%). In a case-control study of 42 former VLBW infants, pCMV positive infants had significantly lower test results at age six years in the simultaneous processing scale of the Kaufman Assessment Battery for Children. In a further follow-up study on 11–16 year old adolescents born at <32 wk GA (19 with, 23 without early pCMV infection), there was evidence of adverse effects of pCMV infection on cognitive function [92]. This was supported by their functional magnetic resonance imaging (MRI) results (n = 15) showing different activations in two brain regions for language performance and differences in grey matter volume compared to children without pCMV infection (n = 19) [93]. Intellectual deficits resulting from pCMV might be more obvious in older children with more complex reasoning. These human data are supported by a neonatal guinea pig model, where postnatally infected pups showed significant cognitive deficits and brain anomalies compared to controls [94].

Notwithstanding the increasing knowledge about short- and long-term consequences of pCMV infection, there is currently no consensus about preventive measures, but further efforts seem justified. The Red Book Committee of the American Academy of Pediatrics recommends serologic screening of mothers of infants born at <32 weeks and to consider short-term breast milk pasteurization in those tested CMV-seropositive [36,37]. Others recommend the pasteurization of breast milk from CMV-positive women in infants born at <28 0/7 wk GA or with a birth weight <1000 g starting on day four until reaching 32 0/7 weeks post-menstrual age [95]. Further prospective studies are urgently needed.

For effectively eliminating CMV from breast milk, heat inactivation is required, whereas freeze thawing is not sufficient. Holder pasteurization (63 °C for 30 min) is safe, but it reduces most of the nutritionally and immunologically relevant components in human milk, such as immune cells, antibodies, enzymes, growth factors and hormones [96]. The authors’ institution therefore practices short-term heat inactivation (heating to 62 °C for 5 s) in their patients born at <32 weeks, as this sufficiently prevents CMV transmission while preserving most benefits of breast milk [81,97].

## 7. Post-Discharge Nutrition

Achieving percentile-parallel growth using mothers’ own milk is the goal of post-discharge nutrition [2]. Breastfeeding of preterm infants, starting with skin-to-skin contact and non-nutritive sucking, should enable predominant breastfeeding at the time of discharge. However, if weaning from the nasogastric tube is impossible, hospital discharge with tube feeding at home and adequate follow-up is the only feasible perspective [98]. In Europe, this applies to approximately 40% of infants <32 weeks GA [99]. A recent meta-analysis of 1251 preterm infants demonstrated that post-discharge formula feeding with 74 kcal/mL does not improve weight or head circumference growth compared to standard term infant formula (≈67 kcal/100 mL) [100]. Limited evidence suggests that feeding preterm infant formula (80 kcal/100 mL, usually in-house available only) compared to standard term formula increases growth rates up to 18 months after birth (mean differences: 500 g weight, 10 mm length, 5 mm head circumference). No convincing evidence exists to support discharging preterm infants with nutrient-fortified mothers’ milk [101]. Therefore, the ESPGHAN guideline recommends individualized post-discharge nutrition adapted to postnatal growth; i.e., for preterm infants with adequate weight gain until discharge, fortified mothers’ milk or a special discharge formula is not required after discharge [2], whereas infants who have grown less well initially should receive fortified breast milk or a special post-discharge formula until at least three months corrected age [2]. This is because of the common observation that former very preterm infants discharged home shortly after discontinuation of naso-gastric tube top-up feeding experience a period of inadequate weight gain [102], which is of unknown clinical importance but at least a major burden for their parents.

## 8. Research Perspectives

Based on the concept that postnatal growth and body composition of preterm infants should ideally mimic intra-uterine growth, and hence that postnatal nutrition should be oriented at placental supply rather than breast milk composition (which is tailored to term infants), we postulate that micronutrients that are actively transported to the fetus against a concentration gradient must be of importance. As an example, fetal plasma concentrations of choline, an essential nutrient for all age groups, are 3–4 times those of the parturient throughout gestation, and very rapidly decrease by 50% or more after preterm birth [54,70]. It is also remarkable that the placenta enriches the fetus with docosahexaenoic acid (DHA) and arachidonic acid (ARA), whereas linoleic acid (LA) is actively held back in the maternal circulation. Based on current feeding regimens for preterm infants, the resulting fatty acid profile of fetal lipoprotein phospholipids (high ARA, low LA, increasing DHA towards term birth) is transformed to adult values (high LA, low ARA, low DHA) within one week. Consequently, the preterm infant’s lipidome at term-corrected age is dramatically different from that of term born infants in all compartments yet investigated, indicating ARA and DHA deficiency and LA-overnutrition [103]. Addressing these and other nutrient deficiencies and imbalances may help further to improve lean body mass growth and long-term outcomes.

## 9. Conclusions

Preterm infants should be provided with all the macro- and micronutrients required to achieve growth as in utero. To minimize side effects of parenteral nutrition, enteral feeding should be started in the first days after birth, preferably based on supplemented mother’s own milk or DHM. Further prospective studies are needed for many aspects of preterm infant feeding.

## Figures and Tables

**Table 1 ijerph-18-07544-t001:** Summary of recommendations for postnatal parenteral nutrition in preterm infants according to the ESPGHAN guideline 2018.

	ESPGHAN Recommendations for Parenteral Nutrition				
Amino acids [21]	- Start: Day 1 with at least 1.5 g/kg/d, day 2 and onwards 2.5 g/kg/d and 3.5 g/kg/d, accompanied by non-protein intakes >65 kcal/kg/d				
Lipids [22]	- Start: (a) immediately or no later than 2 days after birth (b) at discontinuation of enteral feeding at the time of onset of PN - Intake: parenteral lipid intake should not exceed 4 g/kg/day - Essential fatty acids: providing a minimum linoleic acid intake of 0.25 g/kg/day - Administration: continuously over 24 h - Pure soybean oil: May provide a less balanced nutrition than composite intravenous lipid emulsions. PN lasting longer than a few days, pure SO ILEs should no longer be used - continuous over 24 h				
Carbohydrates [23]	- Parenteral glucose supply in mg/kg per min (g/kg per day): Start: day 1: 4–8 (5.8–11.5), day 2 onwards: target 8–10 (11.5–14.4), min 4 (5.8); max 12 (17.3) - Glucose blood glucose levels: Avoid Hyperglycemia (>8 mmol/L/145 mg/dL) or Hypoglycemia (≤2.5 mmol/L/45 mg/dL)				
Calcium, phosphorus and magnesium [24]	Intake in mmol (mg)/kg/d	Calcium	Phosphorus	Magnesium	
	First days	0.8–2.0 (32–80)	1.0–2.0 (31–62)	0.1–0.2 (2.5–5.0)	
	Growing premature	1.6–3.5 (100–140)	1.6–3.5 (77–108)	0.2–0.3 (5.0–7.5)	
Vitamins [25]	- Infants receiving PN should receive parenteral vitamins - Administration of water- and fat-soluble vitamins in fat emulsion to increase vitamin stability - Recommendations for doses of individual vitamins are provided in the guideline [25], but optimal doses and infusion conditions for vitamins in infants are not known				
Trace minerals [26]	- Following trace minerals shoud be provided in preterm infants with PN: - Zinc: 400–500 µg/kg/day - Copper: 40 µg/kg/day - Iodine: 1–10 µg/kg daily - Selenium: 7 µg/kg/day - Manganese: in long term PN max. 1 µg /kg/day - Molybdenum: in long term PN max. 1 µg /kg/day				
Iron [26]	- Prefer enteral rather than parenteral administration - NO administration in short term PN (<3 weeks) - Monitoring of iron status - If necessary: 200–250 µg/kg/day - CAVE: no intravenous iron preparation is approved for pediatric use in Europe				

**Table 2 ijerph-18-07544-t002:** Practice Points.

Parenteral Nutrition	- Bridge the period until full enteral feeding is established [21,22,23,27] - Start immediately after birth with glucose, amino acids and fat [21,22,23] - Use standardized parenteral nutrition solutions for most preterm patients [28]
Transition from parenteral to enteral nutrition	- Start enteral nutrition on day 1 [21] - Accelerate enteral feeding volume in daily increments of 25–30 mL/kg [29] - Ignore gastric residuals as long as abdominal findings are normal [17,30]
Enteral Nutrition	- Promote and support lactation [31] - Bridge the period until sufficient mother’s own milk production with DHM [2,20,31,32,33] - Fortify breastmilk to improve postnatal growth [2,34], consider adapted or individualized fortification to meet nutritional needs [35] - Consider short-term pasteurization in ELBW/VLBW to prevent pCMV infection [36,37]

## Data Availability

Not Applicable.

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
