# Peer review of "Optimizing Early Neonatal Nutrition and Dietary Pattern in Premature Infants"

_ijerph, 2021, doi:10.3390/ijerph18147544_

Round 1

Reviewer 1 Report

The present paper is approaching an essential clinical nutrition issue, the premature child nutrition namely early full enteral nutrition (based on expressed breast milk) on the corresponding 24-34 weeks postmenstrual age - in order to achieve growth rates similar to the in utero growth.

Author Response

Response to reviewers’ comments: Reviewer 1

We sincerely thank you for your helpful comments.

Reviewer #2:

  1. Title

Optimizing Nutrition and Dietary Pattern in Premature Infants

– may include postpartum, or early neonatal nutrition – in order to point the focus of the paper on the postpartum new born nutrition.

We have changed the title accordingly:

“Optimizing Early Neonatal Nutrition and Dietary Pattern in Premature Infants”

Reviewer #2:

  1. Abstract

-  Please review the English language correctness. The context of the abstract should point the fact that the main early postpartum nutrition approach in parenteral so the paper emphasizes the need for total enteral nutrition with expressed breast milk.

 The conclusions are missing

We now have tried to elaborate on the aspect that parenteral nutrition should only be a bridging intervention for a short period of time:

Page 2, line 41-42: “Along with the now well-established early parenteral nutrition, this review emphasizes enteral nutrition, which is started early and can be rapidly increased. To minimize side effects of parenteral nutrition and improve outcome, early full enteral nutrition based on expressed mothers’ own milk is an important goal.”

Reviewer #2:

  1. References

-  Most of reference actual, however, actual research references (not older than ten years) would be welcomed; -  Please check the correctness of references format

We appreciate the reviewer’s desire to have only the current literature summarized in this review. We revised the manuscript and found only few aspects, for which literature older than 10 years had been cited. These are studies on nutrition or protein/calorie intake, which first reported a relationship between nutritional intake and outcome (page 5). Also, some studies on the effects of pasteurization on breast milk (page 13) were >10 y old. In both areas we feel that these are important studies so we did not replace them.

Reviewer #2:

  1. Introduction (Background)
  • -  the knowledge gap is insufficiently mentioned, a better description of the current knowledge regarding the topic would be needed. So, the purpose of the paper better justified and also the research objectives.

Despite various systematic reviews in the field, there is still a lack of evidence on various aspects of neonatal nutrition. This deficiency is now highlighted in the abstract as well as in the introduction as follows:

Page 2, Line 43-46: “Although neonatal nutrition has improved in recent decades, existing knowledge about, for example, the optimal composition and duration of parenteral nutrition, practical aspects of the transition to full enteral nutrition and the need for enrichment of breast milk are still limited and intensively discussed.”

Page 4, Line 82-85: “In the last few decades, an increasing number of studies have been carried out regarding the nutrition of premature infants and new findings have been obtained, but knowledge remains limited in many aspects. Additionally, long-term outcome data in infants following various nutritional interventions are often lacking.”

Reviewer #2:

  1. Materials and methods

- Methodology Major revision

 Paper not approached in the systematic review type according to PRISMA guidelines (http://www.prisma-statement.org/)

This paper is not a systematic review and has therefore not been edited according to the Prisma statement. We performed a literature search in Pub med and the Cochrane Library and added to the introduction:

Page 4:, line 86-88:  “Therefore, we searched PubMed and the Cochrane library for each of the topics addressed above to provide an overview of current evidence, open questions, as well as currently discussed aspects.”

Reviewer 2 Report

Please see attached review, thank you.

Author Response

Response to reviewers’ comments: Reviewer 2

Thank you very much for your review and helpful comments, which helped to improve our manuscript. The new changes in the revised manuscript are highlighted.

Reviewer #2:

Although this is not a systematic review, an important area to address before publication would be the inclusion of some methods to help readers evaluate the evidence summarized (e.g., what databases were searched, what were the selection criterion? [e.g., how were the articles included in this review selected? how were articles included/ excluded?], what is the quality of the evidence?).

We have added our approach to the literature search:

Page 4:, line 86-88:  “Therefore, we searched PubMed and the Cochrane library for each of the topics addressed herein to gives an overview of the current evidence, open questions, as well as the currently discussed aspects.”

Reviewer #2:

The aim of this review is clearly stated, and some rationale is provided for why the authors conducted this review. However, in light of the existing systematic reviews cited by the authors in this manuscript, clearly stating what this review adds above and beyond what is currently published would strengthen this manuscript.

Despite several systematic reviews in the field, there is still often a lack of evidence on various aspects of neonatal nutrition. This deficiency is now highlighted in the abstract as well as in the introduction as follows:

Page 2, Line 43-46: “Although neonatal nutrition has improved in recent decades, existing knowledge about, for example, the optimal composition and duration of parenteral nutrition, practical aspects of the transition to full enteral nutrition and the need for enrichment of breast milk is still limited and intensively discussed.”

Page 4, Line 82-85: “In the last few decades, an increasing number of studies have been carried out regarding the nutrition of premature infants and new findings have been gathered, but knowledge remains limited in many aspects. Furthermore, long-term outcome data in infants following various nutritional interventions are often lacking.”

We have also added:

Page 4, Linge 94-96: “This review summarizes the perspective of a level 3 NICU with a focus on very early enteral nutrition and its repercussions on other aspects of neonatal nutrition”.

Reviewer #2:

Similar to the “practice points” table, other tables summarizing the key messages with references from each feeding domain (e.g., parenteral, enteral, etc.) may help orient readers. It would also be helpful to include references in the “practice points” table.

We have included Table 1 summarizing the recommendations of the current ESPGHAN guideline (2018).

Round 2

Reviewer 1 Report

Dear esteemed authors,

Thank you for your revisions.

Paper quality improved; however, still missing a short conclusion line (as in conclusions) in the abstract.

Also the references format may be slightly revised according to the Journal's requirements.

Kindest regards

Author Response

Response to reviewers’ comments: Reviewer 1

We sincerely thank you for your comments.

Reviewer #1:

Paper quality improved; however, still missing a short conclusion line (as in conclusions) in the abstract.

We added the following sentence to the abstract

Page 2:, line 48-49: “Therefore, further prospective studies on various aspects of preterm infant feeding are needed, especially with respect to effects on long-term outcome.”

Reviewer #1:

Also the references format may be slightly revised according to the Journal's requirements.

The references have now been cited in the stye: "MDPI ACS Journals" as requested in the author guidelines.

Best regards from Tübingen,

Cornelia Wiechers

Reviewer 2 Report

The author’s response to reviewer comments was thorough and thoughtful. I believe that their response addressed reviewer concerns strengthening the manuscript. However, there are a few comments that the authors should address before publication.

Abstract –

  • Some minor copy editing needed (e.g., change “outcome” to outcomes; this sentence is unclear “… breast milk is still limited and intensively discussed”; this is unclear “still missing” [are you referring to existing gaps in the literature?]).
  • Suggest elaborating on the last sentence a bit more – may consider including something about early enteral nutrition since this is the focus of the paper.

Introduction –

  • Sentence starting with “In the last few decades, … number of studies….” needs references.
  • It would strengthen the paper to include a bit more about the current knowledge gap – e.g., briefly describe the current state of the evidence regarding enteral nutrition and highlight how this narrative review adds to and summarizes the literature to inform clinical practice.

Table 1 –

  • It’s unclear what “only as part of clinical trials” means.
  • Should there be units for Ca, Phos, and Mag?

Table 2 –

  • “Fortify breastmilk fortification” – may considering deleting one.

References –

  • Please double check formatting.

Author Response

Response to reviewers’ comments: Reviewer 2

Thank very much you for your comments.

Reviewer #2:

The author’s response to reviewer comments was thorough and thoughtful. I believe that their response addressed reviewer concerns strengthening the manuscript. However, there are a few comments that the authors should address before publication.

Thank you

Reviewer #2:

Abstract

  • Some minor copy editing needed (e.g., change “outcome” to outcomes; this sentence is unclear “… breast milk is still limited and intensively discussed”; this is unclear “still missing” [are you referring to existing gaps in the literature?]).

We changed “outcome” to “outcomes” (line 43).

We rephrased the sentence as follows:
“Although neonatal nutrition has improved in recent decades, existing knowledge about, for example, the optimal composition and duration of parenteral nutrition, or practical aspects of the transition to full enteral nutrition or the need for breast milk fortification is limited and intensively discussed.”

  • Suggest elaborating on the last sentence a bit more – may consider including something about early enteral nutrition since this is the focus of the paper.
  •  

We rephrased the last sentence of the abstract as follows:

This narrative review will summarize currently available and still missing evidence regarding optimal preterm infant nutrition, with emphasis on enteral nutrition and early postnatal growth, and deduce a practical approach.

And at the request of Reviewer 1, we have added the following sentence:Page 2:, line 48-49: “Therefore, further prospective studies on various aspects of preterm infant feeding are needed, especially with respect to effects on long-term outcome.”

Reviewer #2:

Introduction –

  • Sentence starting with “In the last few decades, … number of studies….” needs references.
  • It would strengthen the paper to include a bit more about the current knowledge gap – e.g., briefly describe the current state of the evidence regarding enteral nutrition and highlight how this narrative review adds to and summarizes the literature to inform clinical practice.

We rephrased this paragraph and added important current knowledge gaps as follows:

„In the last decades, an increasing number of studies has been carried out regarding the nutrition of premature infants and new findings have been obtained [16-20], but knowledge remains limited, e.g. concerning the optimal macro- and micronutrient intake through parenteral nutrition (both in the first week after birth and thereafter), the best way to achieve full enteral feeding, or indications for and the optimal composition of a breast milk fortifier“

We have added several important references on different topics.

Reviewer #2:

Table 1 –

  • It’s unclear what “only as part of clinical trials” means.

Thank you. The wording is admittedly misleading, we have deleted it as it does not contain any additional information.

  • Should there be units for Ca, Phos, and Mag?

Previously, we had written the unit for Ca, P, Mg above the table, however, as it may not be easily visible there, we have now inserted it directly into the table.

Reviewer #2:

Table 2 –

  • “Fortify breastmilk fortification” – may considering deleting one.

Thank you! Done.

Reviewer #2:

References –

  • Please double check formatting

The references have now been cited in the stye: "MDPI ACS Journals" as requested in the author guidelines.

Best regards from Tübingen,

Cornelia Wiechers